# Bugs on Drugs: Paracetamol Exposure Reveals Genotype-Specific Generational Effects on Life History Traits in *Drosophila melanogaster*

**DOI:** 10.3390/insects15100763

**Published:** 2024-10-01

**Authors:** Birk Nete Randlev Gleerup Hundebøl, Palle Duun Rohde, Torsten Nygaard Kristensen, Rune Wittendorff Mønster Jensen, Thomas Vosegaard, Jesper Givskov Sørensen

**Affiliations:** 1Department of Biology, Aarhus University, 8000 Aarhus C, Denmark; 2Department of Health Science and Technology, Aalborg University, 9220 Aalborg Ø, Denmark; palledr@hst.aau.dk; 3Department of Chemistry and Bioscience, Aalborg University, 9220 Aalborg Ø, Denmark; tnk@bio.aau.dk; 4Interdisciplinary Nanoscience Center (iNANO), Aarhus University, 8000 Aarhus C, Denmark; rwmj@inano.au.dk (R.W.M.J.); tv@inano.au.dk (T.V.); 5Department of Chemistry, Aarhus University, 8000 Aarhus C, Denmark

**Keywords:** pharmaceutical, insect model, longevity, fecundity, spontaneous activity, epigenetic inheritance, personalized medicine, precision medicine, genotype by environment

## Abstract

**Simple Summary:**

In this study, we used the vinegar fly, Drosophila melanogaster, as a genetic model to investigate the potential long-lasting consequences of exposure to paracetamol—a molecule found in common and widely used over-the-counter medicine for pain and fever relief. We found that increasing doses in the food of developing flies led to increasing mortality. Furthermore, we found that genetically distinct cultures of vinegar flies responded differently to paracetamol exposure when testing life span, spontaneous activity, and production of offspring for two different doses of the drug. These effects were not only present in the individuals directly exposed to the drug but also in the offspring and grand-offspring of these flies, showing the generational effects of the drug. The responses were highly complex and depended on generation after exposure, dose, and individual fly culture. Further investigations into the potentially long-lasting effects of paracetamol exposure and its possible relevance in humans are warranted, as generational effects of pharmaceutical drugs might have consequences for our future generations that are currently overlooked.

**Abstract:**

Few investigations have been made to determine whether pharmaceutical drugs cause any generational effects. These effects can be divided into intergenerational and transgenerational effects. In insects, the F1 offspring of exposed individuals are considered to show intergenerational effects (as they have been exposed as germ cells or early embryos), while the F2 generation is fully non-exposed and considered to show transgenerational effects. Here, the common over-the-counter (OTC) drug, paracetamol, is investigated for genotype-specific responses and effects across generations on three life-history traits: fecundity, longevity, and spontaneous locomotor activity levels in the model species *Drosophila melanogaster*. Seven isofemale *D. melanogaster* lines were exposed to a high and intermediate dose of paracetamol determined by a dose–response curve. NMR investigations verified the long-term presence of paracetamol in the food substrate. Phenotypic effects of paracetamol ingestion were investigated on flies exposed to the drug and in their offspring and grand-offspring. The dose–response curve indicated genotype-specific responses to paracetamol. In the following experiment, all traits investigated displayed significant effects of paracetamol ingestion for at least one of the seven isofemale lines, and we detected strong genotype-specific responses to paracetamol. Fecundity tended to increase in individuals directly exposed to the drug whereas fecundity in the F2 generation was reduced (transgenerational). Longevity generally decreased in directly exposed individuals but tended to increase in F1 offspring (intergenerational). Paracetamol effects on spontaneous locomotor activity were primarily detected as transgenerational effects and were rarely seen in directly exposed individuals. However, across lines, no clear overall trend could be determined for any trait. The generational effects and marked genotype-specific response to paracetamol warrants further investigation of both genotype-specific responses and generational effects in general.

## 1. Introduction

Environmental conditions faced by organisms may have generational effects. Such effects arise when events in one generation have consequences, not stemming from genotypic changes, for phenotypic traits in subsequent generations [1,2]. Generational effects have been documented as a result of exposure to various chemicals [3,4,5,6,7,8,9,10] and environmental changes [11,12,13,14,15,16]. Examples of these span from pesticides, metals, and illicit drugs to temperature and starvation. The phenotypic responses to environmental exposures that are passed across generations are typically divided into intergenerational and transgenerational effects [1,2]. For intergenerational effects, offspring are directly exposed to an effect in utero or as a developing germ cell. Transgenerational effects describe the results of ancestral exposure in fully unexposed descendants. Generational effects (covering both inter- and transgenerational) can be mediated by several different epigenetic mechanisms that, through effects on gene activation, can modify phenotypes [2,17,18,19]. Generational effects encompass both adaptive responses and neutral or detrimental mechanistic consequences of the applied exposure. Regardless, the existence of such generational effects and their underlying mechanisms have potential consequences for understanding a range of questions within ecology, evolutionary biology, risk assessments, and ultimately human health [20].

Paracetamol is a widely used antipyretic and analgesic over-the-counter (OTC) pharmaceutical compound that mainly impacts the central nervous system [21,22,23]. Paracetamol inhibits prostaglandin synthesis [21,22,23] and can generally reduce pain perception [24], lower fever [25], and diminish inflammation [26]. The diverse physiological mechanisms affected by paracetamol lead to a range of direct side effects, but it may also cause generational effects currently overlooked in studies of directly exposed individuals [10]. Such effects are particularly difficult to investigate in humans due to generation time as well as experimental and ethical restrictions. While an effect cannot be assumed to be identical between humans and the vinegar fly model, *Drosophila melanogaster*, this species provides a powerful model to study such effects. The fly model is considered relevant as it has homologous genes for a range of phenotypes relevant to human health and allows for extensive experimentation, phenotyping, and molecular manipulation [27,28]. For example, paracetamol exposure in *Drosophila* resulted in physiological responses comparable to those of mammalian model systems [29]. In *D. melanogaster*, ingestion of paracetamol has also been shown to prolong lifespan [30] and minimize multiple age-related physiological dysfunctions [31,32,33,34]. Specifically, *D. melanogaster* has been proposed as a valuable model for age-related locomotor activity effects [35], and molecular mechanistic comparisons between flies and humans support a functional link to paracetamol overdose and age-related development of Parkinson’s disease-like symptoms [36]. Finally, some reports suggest that fertility in both flies and humans is negatively affected by paracetamol exposure [37,38,39]. Locomotor activity and the age-dependent change in this activity, fertility, and longevity thus all show effects of direct exposure to paracetamol, but it is currently unknown whether effects on these traits are seen across generations.

While our knowledge of the generational effects of drugs including paracetamol is limited, even less is known about variation in responses among different genotypes [40]. The *D. melanogaster* model is excellent for investigating genotype by environment (GxE) interactions, i.e., the variation in the response to environmental conditions among genotypes. GxE interactions are central to many fields of evolutionary and genetics research. It is even a cornerstone for the fast-developing field of personalized medicine, which theorizes that each patient responds differently to medical treatments, and this should be considered when deciding the best course of action to treat a malignant condition [41]. Generational effects are similarly likely to be genotype dependent, but the response to paracetamol exposure and the degree of variation in generational effects among different genotypes is not well studied.

Here, we investigate the phenotypic effects of direct exposure to paracetamol and the generational effects in seven isofemale lines of *D. melanogaster*. Based on suspected side-effects of the drug regarding alterations in fecundity, longevity, and spontaneous locomotor activity (evaluated as a simple proxy for neurological changes), we investigated the effect of paracetamol ingestion on traits in adult individuals fed paracetamol during development (F0) and in adults of two subsequent offspring generations (F1 and F2). In the first experiment, we hypothesized that increased paracetamol concentration in larvae substrate led to progressively decreased egg-to-adult viability. Furthermore, we predicted that an outbred and genetically diverse population would show less sensitivity to paracetamol as compared to an inbred isofemale fly line. In a second experiment, we hypothesized that sub-lethal concentrations of paracetamol would have phenotypic consequences on fecundity, longevity, and spontaneous locomotor activity, with effects being stronger at a higher concentration and more pronounced in directly exposed individuals compared to offspring in subsequent generations. For fecundity, we predicted a decrease in offspring production with paracetamol exposure. The direction of change to spontaneous locomotor activity was not easy to predict but was expected to depend on and be more pronounced with increasing age. For longevity, we expect an increase at low doses of paracetamol but a decrease at higher levels. Finally, based on the difference in genotype composition among the isofemale lines, we hypothesized that each line would display a genotype-specific response to paracetamol exposure.

## 2. Materials and Methods

### 2.1. Study Specimens

For Experiment I (Dose–Response experiment), an outbred *D. melanogaster* population collected at an orchard near Odder, Denmark, was used (coordinates appr. 55.976, 10.176). The population was based on offspring from 25 field-caught mated females and since the collection date (2017), has been maintained in high numbers to reduce genetic drift (>1000 flies per generation) at 19 °C, 12 h:12 h light:dark on a standard *Drosophila* oatmeal-sugar-yeast-agar medium [42]. Before experimentation, we transferred a set of replicate bottles to 25 °C for two generations for temperature acclimation. In Experiment I, we additionally included a single isofemale line. For Experiment II, we used seven lines from a panel of isofemale lines established from the offspring of single-mated females collected from a natural population (Balina, Australia) by Prof. J.S.F. Barker (University of New England, Armidale, Australia) in 2020. All isofemale lines were kept at 25 °C with a 12:12 h light:dark cycle. A standard *Drosophila* oatmeal–sugar–yeast–agar medium was used unless otherwise specified. 

### 2.2. Experiment I: Paracetamol Dose–Response Experiment

A dose–response curve for egg-to-adult viability was created to determine *D. melanogaster*’s response to paracetamol [43] and to select which doses of paracetamol should be used for the generational experiment. We prepared 11 paracetamol treatments at concentrations of 0–100 mM paracetamol (4′-hydroxyacetanalide, Merck, Darmstadt, Germany, CAS-No 103-90-2) dissolved in water, with a 10 mM increase between concentration intervals (Appendix A). For each vial, we added 7 mL of a stock solution to ~2 g of instant Carolina fly food (Formula 4-24^®^ Instant *Drosophila* Medium, Carolina Biological Supply, Burlington, NC, USA). Ten replicate vials were created for each concentration and exactly 50 eggs were added to each vial (see Figure 1). The eggs were collected from food bottles where batches of 50–150 flies were left to oviposit overnight. A similar setup was used for an isofemale line for comparison with five replicates for each dose of paracetamol. The number of eclosing adults was counted for each vial to create a dose–response curve. Concentrations of 20 and 40 mM were used for further experiments as, at a dose of 50 mM, we observed increased development time (unrecorded observation) and a significantly decreased egg-to-adult viability at even higher doses.

To verify that developing larvae were exposed to paracetamol in their substrate, we independently determined the concentration of paracetamol in the substrate by nuclear magnetic resonance (NMR). Eight vials with the dose of 40 mM paracetamol were prepared and left at 4 °C. At various time points (up to 45 days), samples were used for NMR experiments to investigate the temporal stability of paracetamol in the fly diet. The samples were kept at low temperatures to mimic standard storage conditions and to prevent the growth of bacteria and fungi. As this decreased the speed of the potential degradation of the chemical, we extended the duration of the experiment to partly compensate for this. For the NMR samples, eight technical replicates were characterized. Each sample was prepared by weighing out 50–100 mg fly food per vial/time point and suspending it in 900 µL of a solution of sodium trimethylsilylpropanesulfonate (DSS). The DSS solution was prepared by weighing out 10–15 mg of DSS and solubilizing it in 40 mL water. In total, 100 µL of D_2_O was added to the 900 µL solution. The actual quantities are listed in SI. Overall, 500 µL of the suspension was transferred to a 5 mm NMR tube. The ^1^H-NMR spectra were recorded at 500 MHz on a Bruker Avance III spectrometer at 25 °C using water suppression through the zgesgp pulse sequence. All spectra were recorded using a repetition delay of 6 s and 128 scans. The concentration of paracetamol was measured relative to the DSS concentration by taking the ratio of the integrals of the paracetamol methyl peak at 2.14 ppm and the DSS methyl peak at 0.0 ppm. All analyses were made in EasyNMR [44]. A more detailed account of the NMR analysis can be found in the Appendix A. The analysis showed that the amount of paracetamol was constant throughout the 45-day duration of the experiment. While the conditions of this test did not exactly replicate the conditions of the experiment (temperature and metabolically active larvae and microorganisms), we interpret the results as paracetamol being relatively long-lived in the substrate and as justification that developing larvae were exposed to paracetamol.

### 2.3. Experiment II: Generational Effects of Paracetamol

For each of the seven isofemale lines, four replicate vials were prepared with Carolina fly food as described above for each concentration (0, 20, or 40 mM) of paracetamol (4′-hydroxyacetanalide, Merck, Darmstadt, Germany, CAS-No 103-90-2) dissolved in water (Appendix A). Then, 50 ± 5 eggs (collected as described above) were added to each vial to ensure the development in density-controlled conditions. The experiment continued across three consecutive generations, with several phenotypic traits being evaluated in adults of all three generations. Only the juvenile stages of the first generation were exposed to the paracetamol treatment (see Figure 1). The experiment was divided into two temporal blocks, with lines 1–3 represented in the first and lines 4–7 in the second block, respectively. The juvenile survival and adult mortality varied among each line, generation, and trait leading to differences in sample size. The actual number of flies assayed for each case can be seen in Appendix A. 

### 2.4. Fecundity

To investigate fecundity at day five after the eclosion of adults, 60 vials with 7 mL standard fly medium per dose were set up for the directly paracetamol-exposed generation (F0) and 40 vials per dose in the two subsequent offspring generations (F1 and F2, respectively) at the standard maintenance conditions described above. Each vial included two males and one virgin female at the approximate age of five days. Flies were left for 24 ± 1 h to oviposit before removal from the vials. The vials were subsequently maintained at 25 °C until offspring emerged. If the female or both males died during the 24-h period, the vial was excluded from the dataset. The number of eclosed offspring from each vial was counted to determine fecundity and then kept in single-sex groups until five days after eclosion, after which the procedure was repeated to assess the offspring’s fecundity. Line 2 (L2) doses 20 mM and 40 mM in generations F0 and F1, and dose 0 mM in F2 did not produce enough males to fill all of the vials, producing fewer vials overall and causing 5–15 vials to contain only one male. Line 7 (L7) experienced low fecundity for all doses in F1 and F2, decreasing the total number of vials and including only one male per female in most vials due to a shortage of flies. 

### 2.5. Longevity

Up to 80 male individuals used in the fecundity investigation from each line were saved to determine the effect of paracetamol on longevity for all generations and paracetamol doses. Where possible, additional emerged males were used to supplement if numbers fell below 80. These flies were kept in vials containing 3 mL of standard medium with up to ten individuals and tipped to fresh vials every second or third day to avoid medium dehydration. Dead individuals were recorded when flies were transferred to fresh food vials, and surviving individuals were combined to maintain a density of close to ten individuals per vial. This prevented us from using the vial as a unit of replication and led to each male functioning as an individual datapoint. We adopted this approach to accommodate the potentially strong negative effects of low density [45]. Data for Line 1 (L1) experienced data loss in F1 for dose 40 mM, and therefore, we excluded this treatment from the analysis.

### 2.6. Spontaneous Locomotor Activity

Spontaneous locomotor activity of both young and old male flies was quantified for each line, using a subsample of the males from the investigation of longevity. Locomotor activity was assessed by video tracking (see [46]). Young flies underwent assessment on the day or the day after the fecundity assay finished (6–7 days old). After the video recording, the males were returned to the longevity assay. Videos of older individuals were taken after approximately 20% of the males had died, which occurred 19–25 days after eclosion. Noldus Ethovision video tracker version 16 [47,48] was used to track the spontaneous fly locomotion on the videos, noting the distance moved over time (mm/s) at a resolution of 2.5 datapoints per second. Not all samples were perfectly tracked leading to missing information (NAs in the dataset). As quality control, we inspected these samples to determine if it was caused by mortality or inactive flies, which can be missed by the tracking software. In the case of mortality, these individuals were excluded from the dataset. In the case of inactivity of living flies, the moved distance was manually noted as 0 mm/s. Data were averaged across bins corresponding to each minute of the trial. Data from minute 10–35 after initiation of the trail were used to allow the flies to settle down after handling, creating 26 individual points of average speed per minute for each fly for the final analysis. A few recordings stopped prematurely and in such cases, all available data past the 10-min cut off were used. 

### 2.7. Statistics

#### 2.7.1. Experiment I

Data analysis was performed using R version 4.2.1 [49]. The influence of paracetamol dose on egg-to-adult survival was investigated using linear models, with Line as a categorial factor and Concentration as a continuous factor. As the response to paracetamol concentration was strongly non-linear, we fitted the data to a 3rd order model as model reduction (ANOVA function in R) showed a significant loss of explanatory value for simpler models. Concentrations around the drop of the curve (30 mM to 70 mM) were further compared with the control (0 mM) by using Dunn’s test in the ‘rstatix’ package [50], and Holm–Bonferroni adjustment was utilized to account for multiple testing. The concentration of paracetamol measured by NMR in the fly medium, as a function of time (continuous factor), was also investigated by a linear model. Here, a first order (linear) fit described the data well.

#### 2.7.2. Experiment II

Both fecundity and longevity underwent a square-root transformation to improve normality for applying a linear mixed effects model. To investigate the overall effects, we performed an initial analysis in which *Line* was treated as a random factor while *Dose* and *Generation* were treated as independent variables by using the ‘lme4’ [51] and ‘lmerTest’ [52] packages for R. The model used for both the fecundity and longevity assessment was as follows
$$Sqr\left(No.Offspring\right)\;OR\;Sqr\left(Days\right)=Dose\times Generation+(1|Line)$$

For spontaneous locomotor activity, *ID* (individual fly) was added as a random factor and age was added as a variable to the model. The used model was as follows:$$Moved\;Distance=Dose\times Generation\times Age+\left(1|Line\right)+(1|ID)$$

These models all produced relatively low marginal R^2^ values (see Appendix A). To investigate the effects of individual genotypes, we constructed a new model for each trait using *Line* as a fixed variable. In all cases, the model including *Line* as a fixed factor gave lower AIC. The new models were as follows:$$Sqr\left(No.Offspring\right)\;OR\;Sqr\left(Days\right)=Dose\times Line\times Generation$$
Moved Distance=Dose×Line×Generation×Age+(1|ID)

For all three traits, a Tukey–Kramer multiple comparison test was used through the R package ‘multcomp’ [53]. As only within-line and within-Generation comparisons were considered relevant, the lines were investigated separately for each dataset to minimize α-inflation. The age groups were also separated for spontaneous locomotor activity. Holm–Bonferroni adjustment was utilized to account for multiple tests.

## 3. Results

### 3.1. Experiment I: Paracetamol Dose–Response Experiment

Egg-to-adult survival was affected by increasing the paracetamol concentration of the developmental diet (Figure 2). The analysis revealed that a third-order polynomial provided a superior fit to the data (second order versus third order fit: F_(157,159)_ = 38, *p* = $3.3\times 10^{-14}$). This model revealed a significant effect of Concentration (F_(3,157)_ = 221, *p* < $2\times 10^{-16}$), no effect of Line (F_(1,157)_ = 0.03, *p* = 0.86), but a significant Concentration by Line interaction (F_(3,157)_ = 3.7, *p* = 0.01). The interaction effect manifested itself as a line-specific change in sensitivity to paracetamol at low and high concentrations, respectively (see model plots in Figure 2). The Dunn’s test performed on the combined dataset (excluding Line as a factor) uncovered a decrease in egg-to-adult survival at concentrations above 60 mM paracetamol when compared with the control (see Appendix A). 

NMR results showed relatively stable concentrations across time (see Appendix A). The analysis surprisingly showed a very shallow but statistically significant increase in paracetamol concentration over time (F_(1,62)_ = 12.8, *p* = $7\times 10^{-4}$). The slope (model estimate) was 0.3% per day relative to day 0. While the increasing concentration is likely explained by the evaporation of water from the substrate, the results suggest stability of paracetamol in this substrate, where it could readily be detected throughout the experimental period of 45 days (see also Appendix A including Appendix A).

### 3.2. Experiment II: Generational Effects of Paracetamol

The overall analysis of fecundity showed significant differences for all variables and interactions as compared to the control dose in generation F0, though marginal R^2^ was only 4.1% (Appendix A). When visualized, offspring survival seemed to decrease in the control group compared to both doses in generation F0 and increased in F2 (see Figure 3). When Line was treated as a random factor, the analysis of longevity showed statistical significance on at least one level (of Dose, Generation, or interaction) when compared to the control dose in generation F0. However, marginal R^2^ was only 1.5% (Appendix A). No clear pattern for the effect of treatment can be visually discerned for this trait (see Figure 4). Our overall model for spontaneous locomotor activity revealed no effect of Dose but with a complex pattern for other variables and interactions, including interactions with Dose. For this model, marginal R^2^ was 21.8% (Appendix A). Visualized old flies were less active than young flies and were seemingly more so at paracetamol concentrations of 40 mM (see Figure 5). 

In the analysis aimed at investigating line-specific responses, Line was included in models as a fixed factor. For fecundity, the ANOVA table for the model showed that all variables and interactions were significant, and the model explained approximately 25% of data variation (Table 1). The effect on longevity was significant in all interactions and variables except for Dose, and the model explained approximately 35% of data variation (Table 2). All variables and interactions responded significantly for spontaneous locomotor activity, with a marginal R^2^ of 33% (Table 3). The results were highly complex for all traits, with highly specific responses to paracetamol for individual lines (Figure 6). The pairwise comparisons of doses within Line and Generation (and Age for spontaneous locomotor activity) showed significant line-specific responses (Figure 6).

## 4. Discussion

This investigation documents a clear effect of paracetamol on egg-to-adult viability and suggests a hormetic (beneficial) response at low doses of paracetamol to survival in experiment I. The dose–response curves of the two *D. melanogaster* lines used show a different tolerance between the isofemale and wildtype lines. This supports genotype-specific responses in *D. melanogaster*, such as that has been seen for responses to a range of environmental stressors [40,54,55]. Furthermore, the genotype-specific response was complex, with the isofemale line showing lower survival at low and higher survival at high paracetamol doses as compared to the outbred population. This prevented us from making a simple conclusion regarding a higher sensitivity of a given line as otherwise hypothesized and underlines the importance of genotypic variation and genotype-specific responses in this type of study.

While GxE interactions are not uncommon in similar studies of *D. melanogaster* [40,54,55,56], we are unable to discern any underlying mechanism or relate the results to the backgrounds of the lines (isofemale or outbred). Still, the strong reactions to paracetamol exposure in Experiment I motivated the investigation of potential responses in future generations, as seen in Experiment II. Experiment II offered conclusive evidence for the inter- and transgenerational effects of paracetamol.

The latter experiment provided average limited responses to paracetamol in the assayed traits, masking the pronounced line-specific responses. This indicates a considerable genetic component to an organism’s tolerance and physiological response to paracetamol, supporting that the effects of treatment with this drug will differ between individual genotypes [57,58]. To what extent these differences solely rely on the genetic make-up of the lines or could be partly explained by stochastic differences is not possible to determine. For example, we are not able to determine rates of uptake and excretion for each individual larva. While general conclusions are elusive in the current dataset, a few results are worthy of mention. Our data suggest a general trend toward increased fecundity in paracetamol-exposed F0 flies, supporting the hormetic effect seen in Experiment I. While beneficial responses to low doses of otherwise detrimental conditions (hormesis) are well known [59], the beneficial response to low doses of paracetamol exposure contradicts the existing literature, as paracetamol is known to adversely affect sperm production and increase mutation rates [37,38,60]. A possible discrepancy among reproductive traits measured in this study (offspring survival) and other aspects of reproduction, e.g., egg laying and overall fecundity, assessed in other studies might explain these differences.

In the second block (L4–L7), we supplemented flies with additional yeast after the transfer from the instant medium to improve fecundity. All lines experienced sub-optimal conditions on the instant medium in F0, with lower fecundity observed at all concentrations in F0 of the first experimental block (see Figure 3), and this difference may further affect other traits and generations due to epigenetic responses to the dietary restrictions [11,13,14,16]. The effects of paracetamol on other traits remain seemingly unaffected, as only fecundity severely depended on the diet and additional supplements. However, we cannot rule out that interactions between paracetamol and dietary conditions can have an impact on the observed results. In two lines (L6 and L7), an increase in fecundity was coupled with a decrease in longevity. It may be tempting to attribute this to a known trade-off present between these traits [61,62]. However, in the current study, egg-laying was only measured on a single day (6th day after eclosion). The detrimental effect associated with early breeding in males [63] is also mostly discounted since flies were divided into sexes within approximately two days of eclosion. The resources allocated toward fertility across a fly’s lifetime are thereby limited, and the shortened lifespan should solely be attributed to the influence of paracetamol. Beyond the change in yeast supplement, the same protocols were carefully followed for each generation. Still, we observed that the 0 mM controls among generations often showed different values. The existence of substantial unexplained environmental variance is not uncommon, e.g., in quantitative genetic experiments where it is usually quantified (e.g., [64]). Therefore, we do not compare between generations, as the within-generation comparisons are better controlled. The enigma of the sources of environmental variance warrants attention from future studies. For longevity, the most significant results in F0 show a decrease in lifespan due to paracetamol exposure, which contradicts a previous study on paracetamol’s membrane stabilizing effects [30]. The responses occur primarily at the highest concentration. In subsequent generations, the effects are mostly reversed to prolong the lifespan, and this response is restricted to only a few lines. Changes in spontaneous locomotor activity were explained by exposure to the higher concentration, with a tendency for being transgenerational, and young individuals on the high-dose increased activity. Beyond this, we did not observe correlations between generations, nor whether paracetamol generally increases or decreases activity. The fitness effects of changes to activity are difficult to interpret, but neurological changes leading to any alteration to activity profile could likely be considered detrimental [20,36,65,66,67,68]. We highlight that while spontaneous locomotor activity is a convenient and much-used trait to measure, behavior encompasses a very complex set of phenotypes and the results might depend on the exact trait measured. Still, locomotor activity does seem to be directly related to neurology and therefore is a reasonable proxy for this study [69,70].

For all studied life history traits, responses typically either revert or invert from one generation to another and/or among lines. This occurs for all lines, in at least one trait and dose, where lines experience a significantly altered phenotype from one generation to another. Unless the mutational differences significantly constrict fitness through sterility or lethality, deleterious mutations take many generations to be purged from a population [71]. It is assumed the two generations established here are too few in number to revert any potential mutational effects caused by paracetamol. As such, epigenetic mechanisms and their quick response times are a much more likely cause of the changes seen. Paracetamol has been shown to accumulate in the fat body of *Drosophila* during prolonged exposure [29]. This finding might solely be a snapshot of paracetamol levels at the time of death, and the drug is likely excreted over time after exposure stops in surviving flies. Future studies could investigate to what extent paracetamol accumulation explains the effects of exposure between developing larvae and adults and potentially the first-generation offspring.

In conclusion, we find that the effects of paracetamol exposure on mortality in *D. melanogaster* are genotype-specific and that doses above 50 mM paracetamol cause a marked decrease in egg-to-adult viability in a genotype-specific manner. Furthermore, we found highly trait- and genotype-specific inter- and transgenerational effects of paracetamol supporting the existence of genotype-specific phenotypic responses to OTC drugs. The mechanism by which transgenerational effects are transmitted is largely unknown. Further detailed studies may reveal which epigenetic mechanisms may be involved in the transgenerational effects of paracetamol exposure. Histone modifications, miRNA, or potentially methyladenine are likely targets (but not cytosine methylation, as DNA methylation is rare in adult *Diptera*) [17,72,73,74,75].

Our results have potential implications beyond *D. melanogaster* and other insects. While results cannot be easily and directly extrapolated from flies to humans, it is often argued that many fundamental biological processes have partly shared the homologous mechanisms. As such, it begs the question of whether generational effects of OTC drugs occur in humans. Clinical trials ensure the safety of drugs before they become available on the market [76,77,78]. However, current risk assessment methods do not typically determine the effects of a drug on future generations [20], and therefore, such effects are currently overlooked. 

## Figures and Tables

**Figure 1 insects-15-00763-f001:**
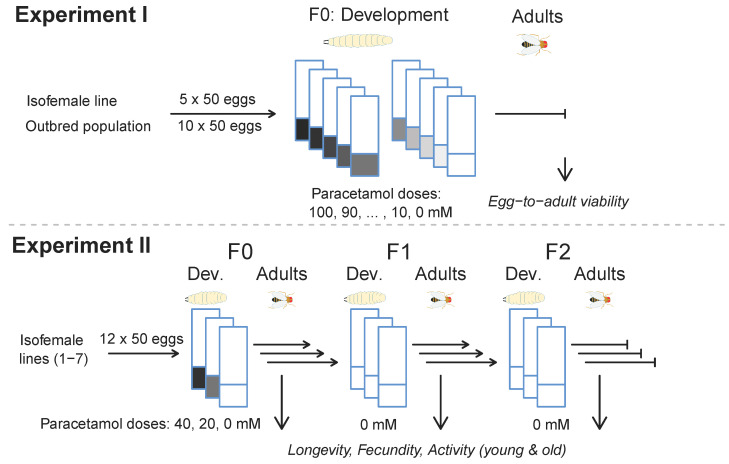
Experimental design of Experiment I (**top panel**) and Experiment II (**bottom panel**). In experiment I, egg-to-adult viability was investigated in five (isofemale line) or ten (outbred population) replicate vials with 50 eggs each.

**Figure 2 insects-15-00763-f002:**
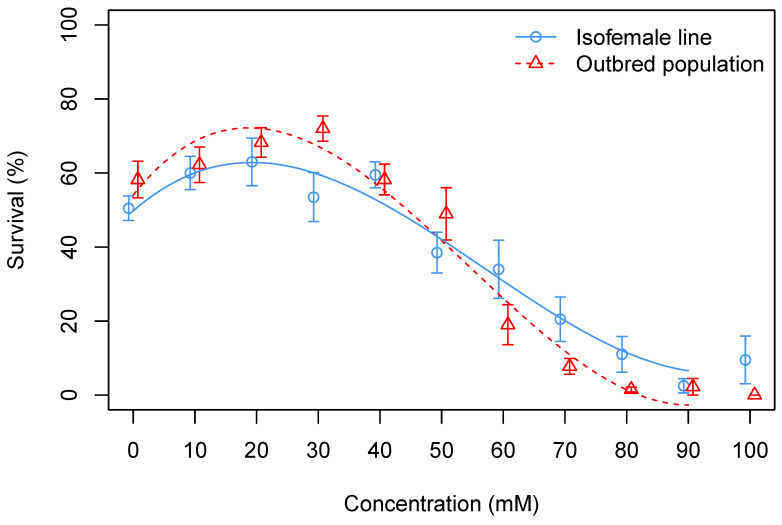
Percentage egg-to-adult survival of individuals from an outbred population (*n* = 10 vials of 50 eggs) or an isofemale line (*n* = 5 vials of 50 eggs) at different concentrations of paracetamol. Points represent means ± standard error of the mean.

**Figure 3 insects-15-00763-f003:**
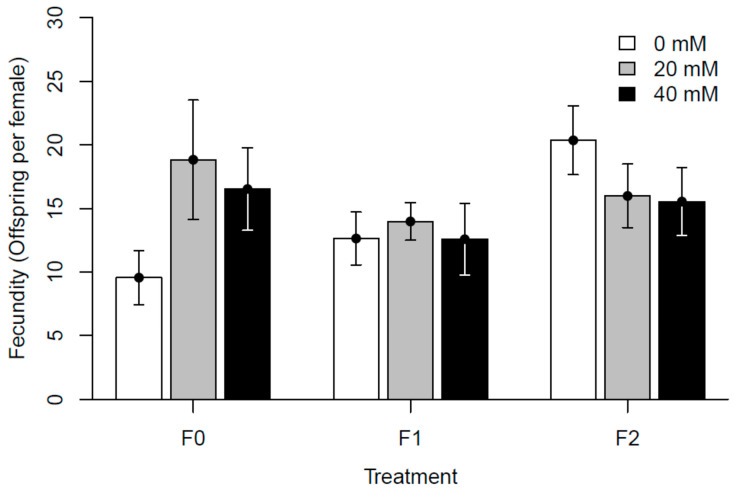
Fecundity of exposed flies (F0) or their offspring across two generations. Bars show the mean of all seven isofemale lines. Error bars show the standard error.

**Figure 4 insects-15-00763-f004:**
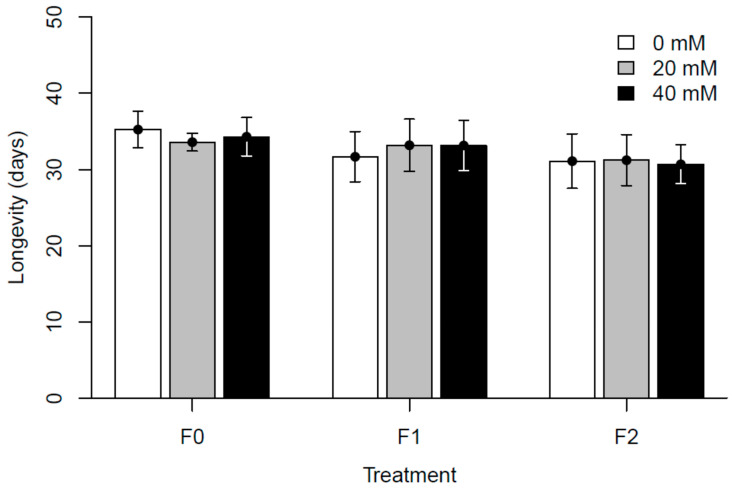
Longevity of exposed flies (F0) or their offspring across two generations. Bars show the mean of all seven isofemale lines. Error bars show standard error.

**Figure 5 insects-15-00763-f005:**
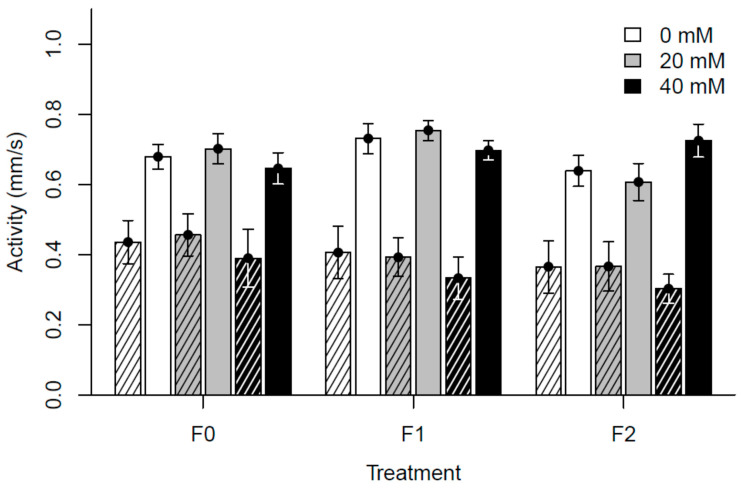
Spontaneous locomotor activity of exposed flies (F0) or their offspring across two generations. Bars show the mean of all seven isofemale lines. Error bars show the standard error. Hatched bars represent old flies and unhatched bars young flies.

**Figure 6 insects-15-00763-f006:**
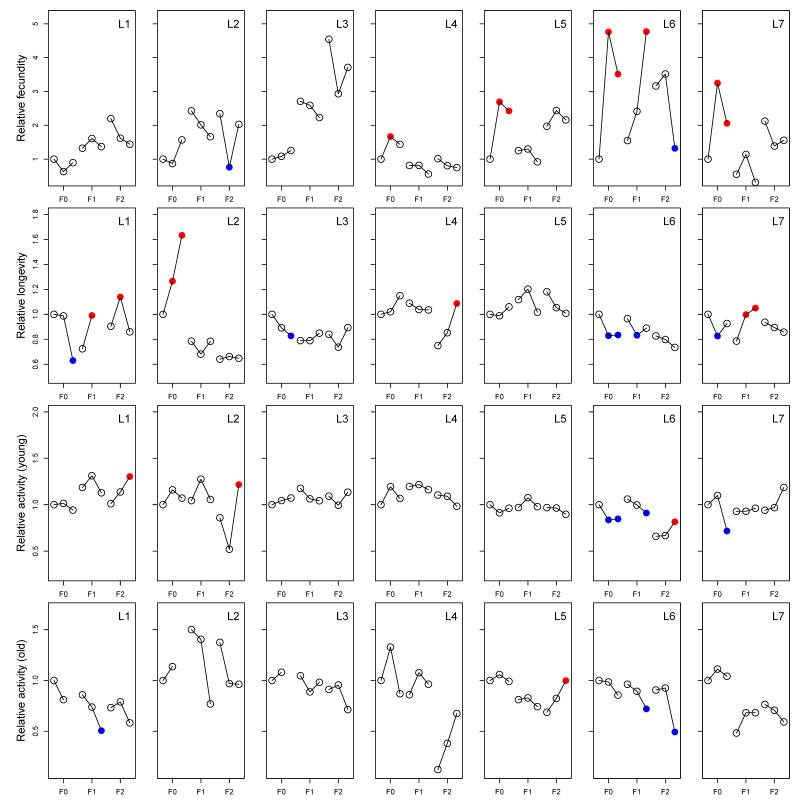
Line-specific responses to paracetamol of seven different isofemale lines (L1–L7) represented in different columns, and for four different phenotypic traits (fecundity, longevity, and young and old spontaneous locomotor activity) represented by rows. Lines show reaction norms to 0, 20, and 40 mM paracetamol exposure in the first (F0) generation. All values are normalized to the F0, 0 mM treatment for all lines for easier comparison of the response to paracetamol. Filled symbols represent significant differences from the associated (same generation) control (red = increased and blue = decreased). For an evaluation of absolute differences between treatments, refer to Figure 3, Figure 4 and Figure 5.

**Table 1 insects-15-00763-t001:** ANOVA table for the fecundity model using Dose, Generation, and Line as independent variables.

Fecundity		SqrSumOffspring			
Predictors	Df	Sum Sq	Mean Sq	F Value	*p*-Value
Dose	2	33	16.67	5.734	0.003
Gen	2	132	65.79	22.628	<0.001
Line	6	694	115.66	39.782	<0.001
Dose:Gen	4	236	58.88	20.252	<0.001
Dose:Line	12	221	18.38	6.323	<0.001
Gen:Line	12	812	67.66	23.271	<0.001
Dose:Gen:Line	24	330	13.76	4.733	<0.001
Observations		2445			
R^2^/R^2^ adjusted		0.262/0.243			

**Table 2 insects-15-00763-t002:** ANOVA table for the longevity model using Dose, Generation, and Line as independent variables.

Longevity		SqrDays			
Predictors	Df	Sum Sq	Mean Sq	F Value	*p*-Value
Dose	2	2.1	1.07	1.576	0.207
Gen	2	72.5	36.27	53.642	<0.001
Line	6	745.4	124.24	183.744	<0.001
Dose:Gen	4	9.6	2.4	3.552	0.007
Dose:Line	12	97.6	8.14	12.035	<0.001
Gen:Line	12	250.6	20.88	30.883	<0.001
Dose:Gen:Line	23	110.5	4.8	7.103	<0.001
Observations		3473			
R^2^/R^2^ adjusted		0.358/0.347			

**Table 3 insects-15-00763-t003:** ANOVA table for the activity model using Dose, Generation, Age, and Line as fixed factors while ID is a random factor.

Spontaneous Locomotor Activity		Distance			
Predictors	Df	Sum Sq	Mean Sq	F Value	*p*-Value
Dose	2	0.152	0.076	4.55	0.011
Gen	2	0.3354	0.1677	10.04	<0.001
Line	6	15.2625	2.5438	152.31	<0.001
Age	1	23.9584	23.9584	1434.56	<0.001
Dose:Gen	4	0.3763	0.0941	5.63	<0.001
Dose:Line	12	0.6415	0.0535	3.2	<0.001
Gen:Line	12	1.1514	0.096	5.75	<0.001
Dose:Age	2	0.189	0.0945	5.66	0.004
Gen:Age	2	0.6102	0.3051	18.27	<0.001
Line:Age	6	4.5025	0.7504	44.93	<0.001
Dose:Gen:Line	24	1.4453	0.0602	3.61	<0.001
Dose:Gen:Age	4	0.2706	0.0677	4.05	0.003
Dose:Line:Age	12	1.0281	0.0857	5.13	<0.001
Gen:Line:Age	12	0.938	0.0782	4.68	<0.001
Dose:Gen:Line:Age	21	2.2027	0.1049	6.28	<0.001
**Random Effects**					
ICC		0.77			
N _ID_		6626			
Observations		171628			
Marginal R^2^/Conditional R^2^		0.329/0.848			

## Data Availability

The original contributions presented in the study are included in the article/Appendix A; further inquiries can be directed to the corresponding author.

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
