# Peer review of "Bugs on Drugs: Paracetamol Exposure Reveals Genotype-Specific Generational Effects on Life History Traits in Drosophila melanogaster"

_insects, 2024, doi:10.3390/insects15100763_

Round 1

Reviewer 1 Report

Comments and Suggestions for Authors

The manuscript presents a compelling investigation into locomotor activity in insects, a topic that is both intriguing and significant for understanding the relationship between neural activity and behavior. The methodology employed for quantifying locomotor activity is robust, providing valuable insights into the effects of various treatments on insect movement. Overall, the results are generally interesting; however, I have several concerns that I would like to discuss further.

Major Points:

  1. Some figures may require clearer labeling or higher quality to enhance understanding. For instance, the blue color in Fig. 1 is overly bright. In Fig. 2, I recommend avoiding the use of only black or gray colors; instead, consider using different colors to highlight key findings. Although some tables include relevant P-values indicating significance, I suggest adding comparative analyses between groups in Figs. 3, 4, and 5, along with the corresponding P-values for significant differences. Please read some relevant papers. Additionally, in Fig. 6, it is essential to highlight key content and use distinct colons to indicate these highlights; otherwise, the authors risk losing clarity in their presentation.
  2. Insects' locomotor activity, particularly in relation to their brain structure and neural activity, varies between spontaneous locomotion and locomotion influenced by infections or other external factors. For example, courtship behaviors in insects differ under various conditions. For further insights, I recommend some relevant papers.
  3. Regarding longitudinal effects, the relationship with insect mortality is not adequately addressed. I understand that statistically analyzing insect mortality can be challenging; however, I still recommend that the authors include results similar to survival curves to provide a clearer picture of the effects over time.
  4. In terms of the interpretation of results, while the writing in the manuscript is generally good, some details lack completeness. For example, in the introduction, I suggest adding more descriptions regarding the research on insect locomotor activity to provide a more comprehensive background.

Comments on the Quality of English Language

Minor Points:

  • The phrase “These effects have been widely recorded in different species and studies” should be revised to “These effects have been widely recorded across various species and studies.”
  • The manuscript alternates between "paracetamol" and "acetaminophen." It is recommended to use the full name upon first mention and then consistently use one term throughout the paper for clarity.

Author Response

Referee 1

The manuscript presents a compelling investigation into locomotor activity in insects, a topic that is both intriguing and significant for understanding the relationship between neural activity and behavior. The methodology employed for quantifying locomotor activity is robust, providing valuable insights into the effects of various treatments on insect movement. Overall, the results are generally interesting; however, I have several concerns that I would like to discuss further.

  • We are happy that the referee appreciates our work and thank for the constructive criticism.

Major Points:

  1. Some figures may require clearer labeling or higher quality to enhance understanding. For instance, the blue color in Fig. 1 is overly bright. In Fig. 2, I recommend avoiding the use of only black or gray colors; instead, consider using different colors to highlight key findings. Although some tables include relevant P-values indicating significance, I suggest adding comparative analyses between groups in Figs. 3, 4, and 5, along with the corresponding P-values for significant differences. Please read some relevant papers. Additionally, in Fig. 6, it is essential to highlight key content and use distinct colons to indicate these highlights; otherwise, the authors risk losing clarity in their presentation.
  • We have revised several of the figures based on the recommendations. We have attempted to improve clarity as suggested, while maintaining a design that still allow the figures to be read in a black/white print. Specifically, we have changed the tone of the blue color in Figure 1, applied red and blue color to lines and symbols in Figure 2. We have revised Figure 6, to reduce the white space, and have increased the font and symbol size. Furthermore, we now indicate if treatments (doses) were statistically different from controls within each generation using filled symbols. In color view, we further indicate by red or blue the direction of the differences (higher or lower, respectively) as compared to the control. Regarding the figures 3-5, we have not made any changes. We have not made any simple comparisons among doses within generation. The overall statistical analysis was quite complex, and we argue in the paper that the huge variation among lines would make overall comparisons and interpretation redundant. Therefore, we proceeded with the line specific analyses and have indicated the result in Figure 6 as explained above.

  1. Insects' locomotor activity, particularly in relation to their brain structure and neural activity, varies between spontaneous locomotion and locomotion influenced by infections or other external factors. For example, courtship behaviors in insects differ under various conditions. For further insights, I recommend some relevant papers.
  • We are unsure what the referee precisely are asking for here. We agree that locomotor activity is a complex trait and under strong environmental influence. We have added a few sentences and some references to the discussion (where we found it suited better) providing a little more detail on the interpretation of the locomotor trait and that behaviors constitute complex and multifaceted traits.
  1. Regarding longitudinal effects, the relationship with insect mortality is not adequately addressed. I understand that statistically analyzing insect mortality can be challenging; however, I still recommend that the authors include results similar to survival curves to provide a clearer picture of the effects over time.
  • We originally included Cox proportional hazard models in the analysis of the longevity data. However, based on AIC scores, these models were inferior (but qualitative similar) to the models applied here and the former were therefore not reported in the results.
  1. In terms of the interpretation of results, while the writing in the manuscript is generally good, some details lack completeness. For example, in the introduction, I suggest adding more descriptions regarding the research on insect locomotor activity to provide a more comprehensive background.
  • Please see the response to point 2 above.

Comments on the Quality of English Language

Minor Points:

  • The phrase “These effects have been widely recorded in different species and studies” should be revised to “These effects have been widely recorded across various species and studies.”
  • Searching the document, supplementary material and cover letter for sentences with resemblance to the one mentioned above (even fragments) was unsuccessful. Clearly, we must be overlooking something and will of course be happy to make the suggested change.
  • The manuscript alternates between "paracetamol" and "acetaminophen." It is recommended to use the full name upon first mention and then consistently use one term throughout the paper for clarity.

Searching the ms for “acetaminophen” only provided hits in the reference list. The chemical name “4’-hydroxyacetanalide” was mentioned once in the materials and methods. Therefore, we are uncertain what should be changed. 

Reviewer 2 Report

Comments and Suggestions for Authors

The manuscript of Rhode et al describes the generational effects of paracetamol treatment in Drosophila melanogaster. For this, the authors investigate three life-history traits in seven isofemale D. melanogster lines. Even though all traits showed effects of paracetamol ingestion in at least one of the lines, no clear overall effect could be determined for any of the analysed traits. The text is well written, the experiments are clearly described and the results carefully presented and discussed.

This reviewer has a general question regarding the use of D. melanogaster as model to study the generational effects of paracetamol treatment. Can it actually provide information about it? A search in PubMed indicates an absence of similar studies and I wonder if it has been attempted before and similar results, that were never published, were obtained. The present study was well designed and the hypothesis were clearly formulated. Considering the drug investigated and the model system employed I understand that its publication is justified.

Author Response

Referee 2

The manuscript of Rhode et al describes the generational effects of paracetamol treatment in Drosophila melanogaster. For this, the authors investigate three life-history traits in seven isofemale D. melanogaster lines. Even though all traits showed effects of paracetamol ingestion in at least one of the lines, no clear overall effect could be determined for any of the analysed traits. The text is well written, the experiments are clearly described, and the results carefully presented and discussed.

This reviewer has a general question regarding the use of D. melanogaster as model to study the generational effects of paracetamol treatment. Can it actually provide information about it? A search in PubMed indicates an absence of similar studies and I wonder if it has been attempted before and similar results, that were never published, were obtained. The present study was well designed and the hypothesis were clearly formulated. Considering the drug investigated and the model system employed I understand that its publication is justified.

  • We thank the referee for the positive and encouraging comments. As pointed to by referee 3, there is a recent reference ‘Saeedi et al. 2022”, which is now included in the introduction to show the value of the Drosophila model in this specific context.

Reviewer 3 Report

Comments and Suggestions for Authors

Summary.

Overall, this is an interesting work on the effects of paracetamol exposure in the larvae stage and the effects in adults in subsequent generations. The authors showed that paracetamol exposure causes alterations in adult success development. Such effects are altered depending on the initial dose. Moreover, the authors showed that such exposure during the larval stage of F0 reaches subsequent generations. The work informs the field about the effects of paracetamol exposure in multiple generations, an aspect that has not been reported. However, the fundamental major flaw of this work is that transference -and presence of paracetamol in active form across generations is not considered. Some significant aspects need to be addressed to improve the paper.

Lines 64-70. The introduction read well, and the authors mention the advantages of the Drosophila system as a model to test drug generational effects. They also highlight the potential of the Drosophila model to get information that can be transferred to humans. The Drosophila model has been used before to test the effects of paracetamol (acetaminophen), Saeedi BJ, Hunter-Chang S, Luo L, Li K, Liu KH, Robinson BS. Oxidative stress mediates end-organ damage in a novel model of acetaminophen-toxicity in Drosophila. Sci Rep. 2022 Nov 11;12(1):19309. doi: 10.1038/s41598-022-21156-w. PMID: 36369211; PMCID: PMC9652370. Saeedi et al. 2022 found that acetaminophen accumulates in the fat body and causes oxidative liver injury. I recommend that such work be cited in the introduction. In addition, it might be useful to improve the discussion section.

Line 263 and line 285. The authors can improve the result section by adding subheadings. Such subheadings should be more informative and state the experiment's take-home message.

Major comments

Line 278 and Supplementary material Figure S1.

In material and methods (Lines 141-142), the authors indicated that paracetamol was determined in the substrate using nuclear magnetic resonance (NMR). The results shown in Figure S1 indicate the relative content of paracetamol. The paracetamol signal in the food vials seems stable over time. However, according to the material and methods section (line 144), food vials were inoculated with paracetamol and left at 4°C. My main concern is that such measurements do not correlate with the experimental conditions in experiments I and II. The variable temperature needs to be considered, and even more importantly, the flies used are metabolically active, which might affect paracetamol degradation in the substrate. In lines 145-148, the authors mention that vials were kept at 4°C to prevent the growth of bacteria and fungi. However, in the experimental conditions, the flies have a natural crawling behavior and a metabolic active microbiome, which naturally involves substrate and bacteria growth changes that might affect paracetamol metabolism. To verify that developing larvae were exposed to stable concentrations of paracetamol across the time in their substrate, the authors need to measure such concentration in the food using similar conditions (temperature and food vials in which larvae are developing).

My second major concern is linked to the first comment. According to Saeedi et al., 2022, acetaminophen accumulates in the fat body. It is possible that larvae exposed to paracetamol during developmental time also accumulate paracetamol in their fat bodies. If this happens, such effects might contribute to an adult’s carry-on paracetamol, probably at lower concentrations than the initial dose, leading to adults chronically exposed to paracetamol molecules. Additional experiments are needed to show if larvae accumulate paracetamol in the fat body and, if so, whether adults still contain such molecules. Such an experiment could be performed using an antibody to detect paracetamol in larvae and - or measuring the concentration of paracetamol in fly samples. Such experiments will show if paracetamol is or is not carried over generations and contributes to the effects observed. Such experiments are necessary to strengthen the impacts observed across generations. Moreover, this observation is important to address because fundamental differences between genotypes can be linked to such paracetamol accumulation in larvae or adults. In that case, even when larvae from different genotypes are exposed to a similar initial concentration (40mM in experiment II), accumulation in the fat body might be different.

Lines 353-354. The authors state that Experiment II offered evidence of the cross-generational effect of paracetamol. However, Figure 6 shows data normalized to the F0 (0 mM) treatment for all lines. Control flies in F1 and F2 (0 mM dose, non-exposed to paracetamol) also show multiple variations compared to F0. Fecundity and relative activity in old flies are the parameters with more variability, mainly between F0 and F2 control flies. This point needs to be clarified or at least discussed.

Minor comments

Table 4. Is “xx” indicating a non-significant result? Please state the meaning of “xx” in the table legend. Also, indicate the meaning of the symbol (-), and indicate if statistical difference corresponds to pairwise comparisons.

Author Response

Referee 3

Summary.

 Overall, this is an interesting work on the effects of paracetamol exposure in the larvae stage and the effects in adults in subsequent generations. The authors showed that paracetamol exposure causes alterations in adult success development. Such effects are altered depending on the initial dose. Moreover, the authors showed that such exposure during the larval stage of F0 reaches subsequent generations. The work informs the field about the effects of paracetamol exposure in multiple generations, an aspect that has not been reported. However, the fundamental major flaw of this work is that transference -and presence of paracetamol in active form across generations is not considered. Some significant aspects need to be addressed to improve the paper.

  • We thank the referee for the insightful and constructive comments.

Lines 64-70. The introduction read well, and the authors mention the advantages of the Drosophila system as a model to test drug generational effects. They also highlight the potential of the Drosophila model to get information that can be transferred to humans. The Drosophila model has been used before to test the effects of paracetamol (acetaminophen), Saeedi BJ, Hunter-Chang S, Luo L, Li K, Liu KH, Robinson BS. Oxidative stress mediates end-organ damage in a novel model of acetaminophen-toxicity in Drosophila. Sci Rep. 2022 Nov 11;12(1):19309. doi: 10.1038/s41598-022-21156-w. PMID: 36369211; PMCID: PMC9652370. Saeedi et al. 2022 found that acetaminophen accumulates in the fat body and causes oxidative liver injury. I recommend that such work be cited in the introduction. In addition, it might be useful to improve the discussion section.

  • We thank the referee for bringing this important paper to our attention. The reference has been included in the introduction. The discussion was revised with the point raised by the referee in mind.

Line 263 and line 285. The authors can improve the result section by adding subheadings. Such subheadings should be more informative and state the experiment's take-home message.

  • The already existing sub-headings were expanded to indicate the nature of each experiment. These has also been added to the comparable sub-headings of the Materials and Methods section.

Major comments

Line 278 and Supplementary material Figure S1.

In material and methods (Lines 141-142), the authors indicated that paracetamol was determined in the substrate using nuclear magnetic resonance (NMR). The results shown in Figure S1 indicate the relative content of paracetamol. The paracetamol signal in the food vials seems stable over time. However, according to the material and methods section (line 144), food vials were inoculated with paracetamol and left at 4°C. My main concern is that such measurements do not correlate with the experimental conditions in experiments I and II. The variable temperature needs to be considered, and even more importantly, the flies used are metabolically active, which might affect paracetamol degradation in the substrate. In lines 145-148, the authors mention that vials were kept at 4°C to prevent the growth of bacteria and fungi. However, in the experimental conditions, the flies have a natural crawling behavior and a metabolic active microbiome, which naturally involves substrate and bacteria growth changes that might affect paracetamol metabolism. To verify that developing larvae were exposed to stable concentrations of paracetamol across the time in their substrate, the authors need to measure such concentration in the food using similar conditions (temperature and food vials in which larvae are developing).

  • We agree that the NMR measurement can’t be expected to correlate one-to-one with the exact concentration the larvae were exposed to in the substrate due to the reasons mentioned. As described in the Materials and Methods section, we extended the duration of storage to partly compensate for the low temperature. We have added the following sentences to the relevant section of Material and Methods: “While the conditions of this test did not exactly replicate the conditions of the experiment (temperature and metabolically active larvae and microorganisms), we interpret the results as paracetamol being relatively stable in the substrate and as justification that developing larvae were exposed to paracetamol.”. We note that the conclusions do not rely on exposure to stable levels of paracetamol, and while interesting, the addition of further experiments is beyond the scope of the current study.

My second major concern is linked to the first comment. According to Saeedi et al., 2022, acetaminophen accumulates in the fat body. It is possible that larvae exposed to paracetamol during developmental time also accumulate paracetamol in their fat bodies. If this happens, such effects might contribute to an adult’s carry-on paracetamol, probably at lower concentrations than the initial dose, leading to adults chronically exposed to paracetamol molecules. Additional experiments are needed to show if larvae accumulate paracetamol in the fat body and, if so, whether adults still contain such molecules. Such an experiment could be performed using an antibody to detect paracetamol in larvae and - or measuring the concentration of paracetamol in fly samples. Such experiments will show if paracetamol is or is not carried over generations and contributes to the effects observed. Such experiments are necessary to strengthen the impacts observed across generations. Moreover, this observation is important to address because fundamental differences between genotypes can be linked to such paracetamol accumulation in larvae or adults. In that case, even when larvae from different genotypes are exposed to a similar initial concentration (40mM in experiment II), accumulation in the fat body might be different.

  • We find the suggestions very inspiring and consider them a valuable next step, which can contribute to pinpointing a mechanistic basis for the observed effects. It is quite likely that accumulated paracetamol in larval fat bodies carries over into the adult stage, which would signify a within generation effect. Developing eggs in these adults could be indirectly exposed by the paracetamol being released from the adult (leading to an intergenerational effect), while it is more likely that responses in the second-generation offspring would signify a true transgenerational effect. We have added a sentence on this topic to the final part of the discussion. Regardless, the main finding of genotype specific responses and carry-over effects of paracetamol exposure is not compromised and therefore we are confident to conclude that the mechanistic approach is beyond the scope of the current study.

Lines 353-354. The authors state that Experiment II offered evidence of the cross-generational effect of paracetamol. However, Figure 6 shows data normalized to the F0 (0 mM) treatment for all lines. Control flies in F1 and F2 (0 mM dose, non-exposed to paracetamol) also show multiple variations compared to F0. Fecundity and relative activity in old flies are the parameters with more variability, mainly between F0 and F2 control flies. This point needs to be clarified or at least discussed.

  • We appreciate this comment by the referee. Ideally, in a controlled experiment like this, the 0 mM controls of each generation would always show similar values. However, environmental variance is typically pronounced in such experiments, even if we try to carefully control everything. That is why we do not compare among generations, as the within generation comparisons are better justified (same batch of food, same generation, same temperatures etc). We agree that this is an important point and have added a few sentences and a recent reference addressing this point in the discussion as suggested.

Minor comments

Table 4. Is “xx” indicating a non-significant result? Please state the meaning of “xx” in the table legend. Also, indicate the meaning of the symbol (-), and indicate if statistical difference corresponds to pairwise comparisons.

  • We agree that the table content was inadequately described. Figure 6 now include the information presented in the table, and therefore we find the table redundant, and it has therefore been removed from the ms.

Round 2

Reviewer 1 Report

Comments and Suggestions for Authors

Thank you to the authors for addressing our concerns one by one, which has mostly resolved our issues. However, we still have a few remaining concerns, including a formatting error in the table between L334-L335 on page 11. Additionally, we suggest uploading the corresponding datasets for each figure to the supplementary files. We also noticed that Table S7, Table S8, and Table S9 were not included in the manuscript description.

Comments on the Quality of English Language

No comments

Author Response

We have re-asked the journal to update the figures and remove the table.

We have re-organised the supplement to link the presented figures to the data available in the supplement.

We now refer to Table S7, Table S8, and Table S9 in the manuscript description.

Reviewer 3 Report

Comments and Suggestions for Authors

Lines 330-333 (Page 11). The Table 4 legend and empty Table 4 are still in the text. 

Lines 341-342. Figure 6 does not show the colors described in the figure legend. 

Author Response

Thanks. We have re-asked the journal to update the ms with deleting the table completely and updating the figures to the new versions.